# Gastrointestinal Permeation Enhancers for the Development of Oral Peptide Pharmaceuticals

**DOI:** 10.3390/ph15121585

**Published:** 2022-12-19

**Authors:** Jae Cheon Kim, Eun Ji Park, Dong Hee Na

**Affiliations:** 1College of Pharmacy, Chung-Ang University, Seoul 06974, Republic of Korea; 2D&D Pharmatech, Seongnam 13486, Republic of Korea

**Keywords:** peptides, oral delivery, permeation enhancers, medium chain fatty acids, sodium salcaprozate

## Abstract

Recently, two oral-administered peptide pharmaceuticals, semaglutide and octreotide, have been developed and are considered as a breakthrough in peptide and protein drug delivery system development. In 2019, the Food and Drug Administration (FDA) approved an oral dosage form of semaglutide developed by Novo Nordisk (Rybelsus^®^) for the treatment of type 2 diabetes. Subsequently, the octreotide capsule (Mycapssa^®^), developed through Chiasma’s Transient Permeation Enhancer (TPE) technology, also received FDA approval in 2020 for the treatment of acromegaly. These two oral peptide products have been a significant success; however, a major obstacle to their oral delivery remains the poor permeability of peptides through the intestinal epithelium. Therefore, gastrointestinal permeation enhancers are of great relevance for the development of subsequent oral peptide products. Sodium salcaprozate (SNAC) and sodium caprylate (C8) have been used as gastrointestinal permeation enhancers for semaglutide and octreotide, respectively. Herein, we briefly review two approved products, Rybelsus^®^ and Mycapssa^®^, and discuss the permeation properties of SNAC and medium chain fatty acids, sodium caprate (C10) and C8, focusing on Eligen technology using SNAC, TPE technology using C8, and gastrointestinal permeation enhancement technology (GIPET) using C10.

## 1. Introduction

Significant attention in the pharmaceutical industry is being given to therapeutic peptides; an increasingly important group of pharmaceuticals [1]. There are currently over 80 peptide drugs in the global market, and research on new peptide therapeutics is ongoing, including over 150 peptides in clinical development and about 600 peptides in preclinical studies [2]. Peptides are a therapeutically unique class of pharmaceuticals poised between small organic molecules and large proteins, with potential as medium-sized therapeutics with properties that differ from those of small chemicals and large biomolecules [3]. Compared with small chemical drugs, peptides can cover a wider area of the target site and thus have superior potency and lower toxicity due to the specificity of interaction with target receptors. Moreover, peptides are often superior to proteins as drug candidates because of their lower cost of production and higher tissue permeability [4,5]. However, a therapeutic drawback of peptide drugs exists; their administration is mainly limited to injectable routes for two major reasons: low gastrointestinal permeability and high enzymatic degradation [6,7,8].

Since injectable peptide drugs require continuous and repeated administration, which causes patient discomfort and has a great effect on medication compliance, pharmaceutical efforts have been directed at improving the invasive administration of injectable formulations [9,10].

The first representative example is the adjustment of the dosing interval through glucagon-like peptide-1 (GLP-1) receptor agonist half-life improvement. Exenatide, first developed in 2005, requires injection twice daily [11]. Since then, products that are administered once daily (lixisenatide and liraglutide) and once a week (dulaglutide, albiglutide, and semaglutide) have been developed by improving the half-life of the peptide. More recently, a daily oral administration product (oral semaglutide, Rybelsus^®^) has been developed [12].

Furthermore, octreotide has also undergone a flow of formulation development to improve dosing convenience. Since the endogenous hormone somatostatin has a short half-life of less than 3 min, octreotide, a synthetic somatostatin receptor ligand with an improved half-life (90–120 min), was developed in the 1980s [13]. Octreotide was initially developed as a subcutaneous injectable formulation administered 2–3 times daily. The frequent administration caused patient discomfort, and in the 1990s, octreotide LAR (long-acting release) product, administered once a month, was developed to improve the dosing interval [14]. Notwithstanding this dramatic dosing interval improvement, the intramuscular injectable form of octreotide LAR required a fairly thick 19-gauge (diameter: 1.1 mm) needle, which caused pain during administration and posed several pharmaceutical problems [15,16,17]. In order to improve the disadvantages of the octreotide injectable administration, an octreotide subcutaneous depot formulation (CAM2029)—with the advantages of being administered once a month, being less painful because of a thinner needle, and its self-administration option by subcutaneous injection—is being developed and is undergoing phase 3 clinical trials [18,19]. Finally, in 2020, the FDA approved an oral octreotide product (Mycapssa^®^) that goes beyond the limits of injectable products (Figure 1).

As seen in the development flow of two peptide drugs, injectable peptide drugs have a history of improved injectable formulations with longer administration intervals and better convenience. Recently, research is ongoing to achieve the final goal of developing oral formulations. Strategies for oral administration of peptides are broadly divided into four methods: (1) permeation enhancers (PEs), (2) nanoparticles [20,21,22,23], (3) lipid-based drug delivery systems (self-emulsifying drug delivery systems (SEDDS) or hydrophobic ion-pairing [24,25], and (4) microneedle devices [26,27,28]. Among these, PEs are the leading strategy. Two hitherto approved oral peptide drugs, Rybelsus^®^ and Mycapssa^®^, have been successfully converted to oral dosage forms using the PE strategy, and research is underway for the development of oral dosage forms of many peptide drugs through a similar strategy.

There are several substances expected to be used as PEs that can be classified into chemical and peptide PEs as per the material standard, and into paracellular and transcellular PEs as per the mechanism standard. Representative examples of chemical PEs that are being applied to various peptides and currently being studied include medium chain fatty acids (MCFAs), Eligen™ technology-based PEs (SNAC, 5-CNAC, 4-CNAB), EDTA, bile salt [29,30], acyl carnitine [30,31], and alkylmaltoside [32] (Table 1). Peptide PEs include peptide-derived microbial toxins like C-CPE [33], AT1002 [34], and angubindin-1 [35], and PN159 [36], an artificial peptide developed using phage display technology. These have clearer mechanisms due to the nature of their origin, albeit with several impending challenges to their practical application, including safety concerns and a sense of greater distance from the classical excipient concept.

Most of the PEs that have entered the clinical trial stage are chemical PEs, and among them MCFAs (C8 and C10) and SNAC, which have secured safety status as a food additive and are generally recognized as safe (GRAS), respectively, and are forefront PEs. In this paper, a brief introduction to the oral peptide drugs (Mycapssa^®^ and Rybelsus^®^) developed using the PE strategy and the development process of PE (MCFA and Eligen™ technology-based PE) applied to each is given, the studies of mechanisms and permeability enhancement are reviewed, and the results of the clinical trial stage are discussed.

## 2. Approved Oral Peptide Drugs

### 2.1. Mycapssa^®^ (Oral Octreotide)

In 2020, the FDA approved Mycapssa^®^, an oral formulation of octreotide used in acromegaly. Chiasma’s Transient Permeation Enhancer (TPE™) technology was used to develop this formulation. Octreotide, a cyclic peptide with a relatively low molecular weight (1019.2 g/mol), was selected as the first candidate for the TPE™ technology application due to its high stability compared to other linear peptides [49,50]. Sodium caprylate (C8) is the key component that acts as a PE in the technology; however, its mechanism as a PE is not clearly elucidated. According to Chiasma, C8 exhibits temporary and reversible permeation enhancement action and induces the reorganization of tight junction (TJ) proteins such as ZO-1 and Claudin, which is presumed to be a paracellular mechanism through TJ modulation [51].

Unlike the unusual case of Rybelsus^®^, which targets absorption in the stomach, Mycapssa^®^ targets drug absorption in the small intestine. Mycapssa^®^ was designed to use an enteric capsule to prevent the breakdown of octreotide in the stomach, otherwise known as drug dissolution, by coating Acryl-EZE^®^ (methacrylate) on gelatin capsules. This capsule contains an oily suspension in which hydrophilic fine particles of octreotide, C8, and polyvinylpyrrolidone (PVP) are suspended in an oil blend containing glycerol monocaprylate and glycerol tricaprylate. Furthermore, to prevent particle aggregation in the suspension, polysorbate 80 in the oil phase was used as a non-ionic surfactant [52]. The dominant PE in this composition is C8; however, it is known that additional PE effects can be expected by using high concentrations of additional additives such as PVP, glycerol monocaprylate, and polysorbate 80 [49].

### 2.2. Rybelsus^®^ (Oral Semaglutide)

Rybelsus^®^ was the first oral peptide drug based on the PE strategy and was approved by the FDA and EMA in September 2019 and March 2020, respectively. Rybelsus, indicated for type 2 diabetes mellitus (T2DM), is an oral formulation of semaglutide with sodium salcaprozate (SNAC) as the PE. SNAC is a synthetic N-acylated amino acid derivative of salicylic acid, developed by Emisphere’s Eligen™ technology [53]. This technology conducted carrier libraries to develop carriers that increase hydrophobicity through non-covalent interactions with the payload (drug), which aims to increase transcellular absorption [54]. Eligen B12 (vitamin B12), approved as an FDA medical food in 2015, was the first product to be commercialized using SNAC. Subsequently, Novo Nordisk transferred Eligen™ technology from Emisphere and applied it to the development of an oral GLP-1 receptor agonist. The precedent approval of Eligen B12 and the safety status of FDA GRAS of SNAC secured in the process made the development of the current Rybelsus^®^ product easier [55].

The contribution to the birth of oral semaglutide products has two main categories, the first of which is chemical modification of peptides. Novo Nordisk successfully launched Victoza^®^ (Liraglutide) by improving the short half-life, which was a significant drawback of GLP-1 and exendin-4 [56,57,58]. This was made possible by acylation (palmitoylation (C16) to Lys-28), and the half-life was improved through enhanced binding with albumin and resistance by dipeptidyl peptidase-4 (DPP-4) [59]. Semaglutide (a peptide) further improved the half-life of liraglutide. In the liraglutide structure, octadecanoic (C18) diacid was added to the acylation of Lys-28 by a bis-aminodiethoxyacetyl linker to acylate it. Moreover, Ala-8 was substituted with an artificial amino acid, 2-amino-isobutyric acid (Aib), preventing decomposition by DPP-4. Through this, it was possible to develop oral formulations (Rybelsus^®^) as well as subcutaneous injectable formulations, which are given once a week (Wegovy^®^, Ozempic^®^) [60].

The second contribution is the improvement of oral absorption by SNAC. Unlike SNAC’s mechanism presented by the existing Eligen™ technology, Novo Nordisk asserts that SNAC increases the transcellular absorption of semaglutide by elevating the local pH around the tablet in the stomach. In other words, it protects against pepsin by increasing the pH near the semaglutide that is eluted after the tablet sinks near the lower mucous membrane of the stomach, improves the solubility of the semaglutide, and induces monomerization in the local environment [61]. According to Novo Nordisk, these PE mechanisms of SNAC work specifically for semaglutide [55].

## 3. Permeation Enhancers–MCFAs (Medium Chain Fatty Acids)

Medium chain fatty acid (MCFA) is a naturally derived fatty acid with a length of about C6 to C12. For about 30 years now, active research has been performed on MCFA as a PE for hydrophilic, impermeable drugs. The Japanese Nishmura and Kitao research team, who obtained patents in the United States in early 1980, devised the application of this series of compounds to PEs [62]. They tried to find a PE that improves drug absorption in suppository form in the rectum by screening carboxyl group derivatives, with the main drug target being beta-lactam antibiotics. Their proposed mechanism was carboxylic acid derivatives increasing drug permeation by opening TJs while temporarily removing calcium ions necessary for maintaining TJs. During this study, sodium caprate (C10) and C8 were selected as the most potent PEs (particularly C10) [63] (Figure 2). Based on this study, ampicillin suppositories using C10 as a PE were approved in Japan and Sweden (Doktacillin™) [64].

Beyond the application of suppositories, MCFA PE studies centered on C8 and C10 have led to oral administration studies of low permeable drugs. Until now, the mechanism of action of C8 and C10 remains poorly defined. However, C8 and C10 are known to act as PEs through a common pathway; they act mainly in the form of a monomer because their action is based on a surfactant-like action. Therefore, the point of focus in these MCFAs is their critical micelle concentration (CMC) [65], because a high CMC ensures the existence of a high concentration of monomeric surfactant that can act as a PE. It is challenging to characterize the CMC of surfactant because it is greatly affected by temperature, ionic strength, pH, and the measurement method [66]. Among MCFAs, C8 and C10 are PEs with high CMC values. Hossain MS et al. conducted a study to determine the CMC of MCFA in various solvent environments through the classical surface tension measurement method (Wilhelmy method) and coarse-grained molecular dynamics (CG-MD) simulation [66]. The experimental results in the environment most similar to the physiological buffer (pH 7, 140 mM NaCl) are as follows. In the Wilhelmy method, C8 and C10 showed CMC values of 42.1 mM and 15.5 mM, respectively. In CG-MD simulation, it was predicted that C8 and C10 have CMC values of 13.88 mM and 8.27 mM, respectively. On the other hand, the calculated CMC values of C12 and C14 with longer carbon chains were 1.09 mM and 0.83 mM, respectively.

The relatively low regulatory hurdles based on the accumulated safety data of MCFAs make them desirable PEs. Both C8 and C10 are FDA-approved food additives [67]. The European Food Safety Authority (EFSA) also concluded that there are no safety problems in using fatty acids ranging from C8 to C18 as food additives. Their rationale is that fatty acids used as food additives are absorbed and metabolized in the same way as free fatty acids from lipid molecules present in the general diet, and although the toxicity database is limited, no sub-chronic toxicity and no genotoxicity have been observed [68].

### 3.1. Sodium Caprylate (C8)

C8 is an eight-carbon saturated fatty acid belonging to the class of MCFAs with a molecular weight of 166.19 g/mol and a pKa of 5.19. It is present in the milk of many mammals, and along with C10 is particularly high in goat milk [69]. Studies on the mechanism of action of C8 as a PE have been mainly conducted with C10 as the MCFA [70,71,72], and C8 alone was dealt with in the study of the TPE™ technology applied to Mycapssa^®^ [51]. In the 1980s, studies on the application of C8 were conducted for rectal suppositories and nasal applications. In these studies, the opening of TJs by Ca^2+^ chelation was suggested as a mechanism of C8 [63,71]. In the 1990s, studies on the effect of C8 and C10 on different molecular sizes and diffusion coefficients, from urea to PEG 900 and inulin in the colon of rats, were conducted to elucidate the mechanisms of C8 and C10 [70]. They argued that the permeability enhancement of C8 and C10 is mainly achieved through the paracellular pathway, which is divided into the shunt pathway (large pore pathway), that is not limited by the molecular size, and the small pore pathway that is limited by the molecular size.

However, these early studies lack evidence to support their asserted mechanism. A study on the permeation enhancement mechanism of TPE™ formulation was conducted in the TPE™ formulation study, which successfully led to the FDA-approval of oral octreotide with a formulation containing C8. This study suggested a paracellular pathway by TJ opening through the reorganization of ZO-1 and claudin proteins [51]. Contrary to the paracellular mechanism asserted in early C8 and TPE studies, recent studies have highlighted evidence for the transcellular pathway in the mechanism of MCFAs including C8 [72], thus classifying C8 as a transcellular PE with surfactant-like action [65,73].

Most studies on the PE effect of C8 have been conducted with C10. A common finding in many studies is that C8 is less potent than C10. In terms of surfactant PE, C8 has a higher CMC value than C10 and has faster membrane-inserting kinetics [72]. Nevertheless, C8 has a lower permeation-enhancing potency compared to C10, probably due to its insufficient chain length because it is composed of carbon numbers close to the threshold of MCFAs that have a PE effect [72,74]. According to a recent molecular dynamics simulation study, since C8 has a greater membrane expulsion tendency after being inserted into the membrane than C10, the ratio of the remaining C8 inserted into the membrane is small, and is thought to reduce the membrane disruption effect [75].

C8 is considered a key PE of the TPE™ technology that has led to the successful launch of Mycapssa^®^. Thus far, only TPE™ has entered clinical trials using C8 as a PE. When the TPE™ formulation was applied to a fluorescent marker (FITC-labeled dextran of 4.4 kDa, FD4) in the intrajejunal administration rat model, it was shown that the area under the curve (AUC, 0–90 min) value increased by over 10 times. In addition, in the absorption enhancing effect experiment, according to the increase in molecular weight of FITC-dextran (4.4–70 kDa) absorption enhancement was shown to be up to 40 kDa, and insignificant absorption enhancement was detected at 70 kDa. This is the result of absorption enhancement of the TPE™ formulation (oily suspension containing C8), which is easily overinterpreted as the permeation-enhancing effect of only C8. In effect, compared to the permeation enhancement of the combination of the FD4 and TPE™ formulation, FD4 and C8 in saline solution showed a relative value of one-fifth [51].

### 3.2. Sodium Caprate (C10)

C10, also called sodium decanoate, is a sodium salt of a saturated fatty acid composed of 10 carbons, with a molecular weight of 194.25 g/mol and a pKa value of 4.95. C10 is a representative MCFA PE and there have been active studies on its ability to enhance the permeability of various drugs, from low permeable chemical drugs [76,77,78,79] to antisense oligonucleotides and peptides (Table 2) [80,81].

Various methods and experimental models have explored the mechanism of C10 as an MCFA PE (Table 3). The mechanism suggested by the first study that utilized MCFAs as a PE was that MCFA opened TJs by chelating intercellular Ca^2+^ [63]. However, several subsequent studies on the mechanism of action of C10 have denied the Ca^2+^ chelating mechanism. According to Anderberg et al. [97], the presence or absence of Ca^2+^ in the apical (donor) part of the Caco-2 cell monolayer permeability assay did not affect the permeation-enhancing effect, and there was no change in cell integrity when Ca^2+^ was removed from the apical part. Rather, Ca^2+^ showed a negative effect by precipitating C10 in the buffer. In the study of Tomita et al. [98], the mechanism of action of C10 was explored through comparison with ethylenediaminetetraacetic acid (EDTA), a representative Ca^2+^ chelating agent. When EDTA was added to the basolateral (acceptor) part. rather than the apical part of Caco-2 cell monolayer permeability assay, the permeation effect was superior. Conversely, when C10 was applied to the apical and basolateral parts, there was no difference between the two. In this respect, the permeation-enhancing mechanism of C10 by Ca^2+^ chelation was denied.

Subsequent mechanistic studies have mainly focused on TJ modulation by C10 treatment and its intracellular pathway. Tomita et al. [38] found that C10 increased intracellular Ca^2+^ and that the permeation enhancing effect of C10 was inhibited when treated with a phospholipase C (PLC)-calmodulin (CaM) signaling inhibitor. However, in a subsequent study of the impact of C10 on airway epithelial cells, Coyne et al. [100] suggested that C10 induces an increase in intracellular Ca^2+^, albeit that the PE effect of C10 is Ca^2+^-independent. They observed the effect of C10 when intracellular Ca^2+^ was removed with an intracellular Ca^2+^ chelator; increased intracellular Ca^2+^ was maintained by treatment with a Ca^2+^-ATPase inhibitor and after treatment with the PLC-CaM signaling inhibitor. In all cases, there was no effect on the reduction of C10 transepithelial resistance, and the redistribution of the TJ protein by C10 was also not affected by the removal of intracellular Ca^2+^.

While the initial study focused on the TJ observation through transmission electron microscopy [48] and the change in TJ proteins through immunofluorescence, active research on the effect of C10 on lipid membranes based on membrane perturbation or disruption is underway [101,104,105]. These studies focus more on the interaction with the lipid membrane, whereby C10 is a surfactant that successfully inserts into the membrane. Sugibavashi et al. [101] investigated the mechanism of C10 using a lipid raft model, which is relatively robust due to the enrichment of cholesterol and sphingomyelin and has resistance to non-ionic surfactants such as Triton X-100. When comparing C10 and Methyl-β-Cyclodextrin (MβCD), which disrupts the lipid raft by depleting cholesterol, MβCD displaced claudin-1,-4,-5 and occludin, while C10 displaced claudin-4,-5 and occludin from the lipid raft. Brayden et al. [104] suggested that the underlying mechanism of C10 could be attributed to membrane perturbation via misoprostol pretreatment. Misoprostol is a synthetic PEG_1_ methyl ester analog that protects gastric and duodenal mucosal perturbation-, NSAID-, aspirin-, and alcohol-induced ulcers [106]. Pretreatment with 10 nM misoprostol for 30 min significantly reduced the PE effect of C10 in both in vivo and ex vivo experiments, and this reduction effect was negated by SC51322, a prostaglandin EP-1 receptor antagonist. In addition to the permeability of the marker, the intracellular Ca^2+^ and mitochondrial membrane potential increase—indicators of the mechanism of action of C10—also showed the same trend with misoprostol and SC51322, respectively. More recently, there have been ongoing in silico studies to predict the molecular dynamic interaction between surfactant-based PEs such as C10 and the lipid membrane. This is an experimental technique that can further characterize C10 membrane perturbation by computing the insertion fraction of C10 into the membrane on the micro-second level [72] and predicting the behavior after membrane insertion, i.e., flip-flop or migration from the membrane to the aqueous phase [75]. However, since it is currently simulated in a simple virtual membrane composed of 1-palmitoyl-2-oleoyl-sn-glycero-3-phosphatidylcholine (POPC), evaluating the effect of the perturbation of such a membrane on TJ proteins or intracellular changes by C10, including an increase in intracellular Ca^2+^, is challenging.

Research on the mechanism of C10 has yielded varying results and hypotheses for TJ modulation and membrane perturbation (Figure 3). However, there remains the possibility of a complex mechanism of C10 that is not among the two established categories, and the definitive mechanism remains unknown. Early research on the mechanism was focused on TJ modulation, but subsequent studies put weight on membrane perturbation according to the surfactant structure of C10. Paracellular routes by TJ modulation and transcellular routes by membrane perturbation are often regarded as two separate mechanisms. However, in a recent study by Brayden et al., a PE exhibiting a mild surfactant structure showed that membrane perturbation was its fundamental mechanism, suggesting the possibility of association with TJ modulation [47,105]. This means that TJ modulation and membrane perturbation may not be independent or completely dichotomous mechanisms, and a link between the two may exist. Through this new insight, a new direction was suggested for the study of the mechanism of C10, whose exact mechanism is unknown.

C10 has been applied in in vitro and in vivo experiments to enhance the permeability of various peptides, ranging from small peptides with a molecular weight of 300 Da to peptides with a molecular weight of 7 kDa (Table 2). In an in vitro experiment (Caco-2 cell monolayer permeability assay), C10 was principally used at concentrations of 2.5–25 mM, which is a concentration range considered for cell viability and the CMC value of C10. The peptide permeability enhancement ratio of C10, which does not include the pharmacodynamic parameters of C10, was shown to be increased by up to 10 times.

The development of oral peptide drugs using C10 as a permeation enhancer started in earnest with the Gastro-Intestinal Permeation Enhancement Technology (GIPET™) of Merrion Pharmaceuticals [107]. It has consistently demonstrated the efficacy of solution-state peptides and C10 via in vitro and in vivo experiments. However, for the actual development of an oral peptide drug to which PEs are applied, a solid dosage form must be implemented. Moreover, in the final formulation, decomposition by gastric acid and simultaneous or optimized release of C10 and peptides at the absorption site should be considered. The relevance of GIPET™ lies in their performance in studies of the design of an actual solid formulation of C10 as a PE. GIPET™ has been developed and studied in three versions of the formulation so far, and the exact composition of the exact components and concentrations of the formulation has not been fully unveiled.

GIPET™ 1 formulation is a solid formulation in the form of an enteric-coated tablet containing C10. GIPET™ 2 contains C8 and C10 mono/diglycerides, and the solid formulation is in the form of an enteric-coated soft gel/hard capsule shell. Information on GIPET™ 3 has not been disclosed [108].

In a patent embodiment using fondaparinux as an active pharmaceutical ingredient (API), it is known that GIPET™ 1 (High) contains 550 mg of C10 and GIPET™ 1 (Low) contains 275 mg of C10 per 5 mg of fondaparinux [109]. GIPET™ 2 is a C8 and C10 microemulsion formulation esterified with glycerol, known as Capmul^®^. GIPET™ 2 Form 1 is a formulation based on Capmul^®^ MCM (glyceryl caprylate caprate) and is composed of 41.9% Capmul^®^ MCM, 23.3% PEG 400, 21.3% Tween^®^ 80, 5% Captex^®^ (glyceryl tricaprylate caprate), and 0.5% (5 mg) of fondaparinux. GIPET™ 2 Form 2 is a formulation based on Capmul^®^ MCM C10 (Glyceryl caprate), and consists of 36.6% Capmul^®^ MCM C10, 23.3% PEG400, 21.3% Tween^®^ 80, 13.3% Captex^®^ 300, and 0.5% (5 mg) of fondaparinux. The pharmacokinetic (PK) parameters of these four GIPET™ 1,2 formulations were measured using a dog’s intra-duodenal instillation model, and compared with 1 mg subcutaneous formulation. Results of GIPET™ 1 (High) 16.94%, GIPET™ 1 (Low) 11.09%, GIPET™ 2 (Form 1) 18.48%, and GIPET™ 2 (Form 2) 9.61% were shown. Furthermore, compared to the unenhanced administration group to which permeation enhancer was not applied, GIPET™ 1 (High) increased by 3.2 times, GIPET™ 1 (Low) increased by 2.1 times, GIPET™ 2 (Form 1) increased by 3.4 times, and GIPET™ 2 (Form 2) increased by 1. 9 times.

Clinical trials have applied the GIPET™ technology to several peptides and proteins (Appendix A). According to a comprehensive paper on the results of phase 1, GIPET™ 1 has an oral bioavailability of 8.4%, 9.0%, and 8.0% for alendronate, LMWH (MW 4400), and LMWH (MW 6010), respectively. Desmopressin, to which GIPET™ 2 was applied, showed an oral bioavailability of 2.4%. Alendronate and desmopressin showed 5-fold and 13-fold increased absorption compared to oral control, respectively [108,110]. In a clinical study of GIPET™ application to acyline, a GnRH antagonist peptide, 10 mg, 20 mg, and 40 mg of acyline were administered to subjects. All three concentrations of GIPET™-enhanced oral acyline decreased the serum follicle-stimulating hormone, luteinizing hormone, and testosterone for almost 12 h. Serum acyline concentration increased immediately after oral administration, but it was difficult to observe differences in pharmacokinetics parameters by dose due to large variability between subjects [111]. Merrion pharmaceutical’s GIPET™ technology has entered into a license agreement for the development of Novo Nordisk’s oral GLP-1 agonist and oral insulin. In particular, it has been applied to the development of oral insulin formulations and has entered clinical trials; however, it is known to have been discontinued [112]. Biocon’s insulin tregopil project is currently developing oral insulin through C10. Insulin tregopil is a novel PEGylated insulin with a structure in which a short PEG is attached to the ε-amino group of Lysβ29 of human insulin [113]. According to the development paper of insulin tregopil, it is a simple, uncoated tablet formulation containing 5–15 mg of insulin tregopil, C10 as a PE, and mannitol, PVP, colloidal silica, and magnesium stearate as the remaining excipients. This tablet targets drug absorption in the stomach and is designed to elute more than 75% of the drug within 15 min [114].

C10, which has been extensively studied as a PE, has positioned itself as a representative MCFA PE. Recently, in addition to classic tablets or enteric capsules, there are ongoing studies on the application of next-generation formulation, such as 3D printed capsules that break under pressure in the antropyloric region and microcontainers that unidirectionally release drugs and PE [115,116].

## 4. Permeation Enhancer: Eligen™ Technology-Based PEs

Unlike naturally occurring MCFAs, Eligen™ technology-based PEs are a family of compounds developed for permeation enhancement. Eligen™ technology, based on PEs developed by Emisphere, has undergone a development process. They originally targeted the oral delivery of peptides through microspheres composed of thermally condensed α-amino acids [117]. In that process, it was necessary to develop a hydrophobic α-amino acid with a low molecular weight for microsphere preparation. They chose a method of derivatizing soy protein hydrolysate with phenylsulfonyl chloride, and through this, they successfully manufactured microspheres. In subsequent permeability experiments, empty microspheres were found to enhance the intestinal absorption of peptides loaded into microspheres. Through additional experiments, they discovered that (phenylsulfonyl)-α-amino acids themselves had the effect of a PE, and through this, they began screening for the PE activity of the modified α-amino acids themselves, and not through the microsphere strategy [53]. This initial study concluded that there is insufficient evidence that the increase in peptide absorption occurs by classical mechanisms such as protease/peptidase inhibition and penetration enhancement, and the research team focused on the possibility of enhancing permeation by specific interactions between peptides and Eligen™ technology-based PEs. Based on the flow of these studies, it can be observed that they were not limited to α-amino acids, but continuously screened for the difference in the permeability enhancing effect according to the changes in the substituents of N-acylated-non-α-amino acid, lipophilicity, etc. [118]. Subsequently, Emisphere’s research team conducted a study on the mechanism of PE action with candidates obtained by screening (not including SNAC, the current leading compound). They argued that their PE acts as a “carrier” for protein/peptide delivery and that the passive/transcellular pathway is the main pathway (Figure 4). More specifically, their carriers stabilize the partially unfolded conformer of protein/peptide through non-covalent bonding, which exposes the hydrophobic side chain of protein/peptide and increases solubility in lipid membranes [119].

Through this journey, SNAC (N-[8-(2-hydroxybenzoyl)amino] caprylate) was developed, which also led to the development of oral semaglutide and other leading compounds such as 5-CNAC (8-(N-2-hydroxy-5-chloro-benzoyl)-amino-caprylate) and 4-CNAB (N-(4-chlorosalicyloyl)-4-aminobutyrate) (Figure 5). The most advanced of these is SNAC, which has been applied in clinical trials with insulin [121], heparin [122,123], ibandronate [124], and peptide YY3-36 (PYY3-36) [125], and led to the successful development of oral vitamin B12 and semaglutide products. The remaining leading compounds are 5-CNAC and 4-CNAB. The compound 5-CNAC has been applied to salmon calcitonin and has moved to phase 3 clinical trials [126]. The compound 4-CNAB, for its part, has been applied to insulin, and phase 1 clinical trials have been completed [127].

SNAC (N-[8-(2-hydroxybenzoyl)amino] caprylate), also called sodium salcaprozate, has a pKa value of 5.01 and a molecular weight of 301.31 Da [128]. It shares structural similarity with MCFA in that it has a fatty acid moiety, but unlike MCFA, it does not adequately demonstrate the tendency of surfactant-like action membrane insertion/perturbation. SNAC has a salicylamide structure at the molecular terminal, in addition to the carboxyl group of fatty acid shared with MCFA, and has a larger distribution of hydrophilic groups. In effect, the computed topological polar surface area of SNAC is more dispersed, with it being 40.1 Å for C10 and 89.5 Å for SNAC [129]. In a recent in silico-based research model, the tendency of SNAC to disrupt the membrane was calculated to be less than that of MCFA due to the presumed salicylamide structure. Moreover, the tendency of expulsion from the membrane leaflet after insertion into the membrane was also greater [75].

There have been studies on SNAC, a representative material of Eligen™ technology-based PEs, for the oral administration of insulin, octreotide, etc. (Table 4). This has led to the successful development of Rybelsus^®^ (oral semaglutide). Most studies on the mechanism of SNAC have agreed that it acts as a transcellular PE (Table 5). Furthermore, many studies have concluded that SNAC does not exhibit surfactant-like action and membrane perturbation tendencies, as MCFA does. The representative mechanism of SNAC is the carrier mechanism. That is, SNAC forms a non-covalent complex with a drug to increase lipophilicity and membrane permeability, which is the mechanism claimed by Emisphere, who developed SNAC [54,130]. The PE mechanism of SNAC cannot be generalized to a single drug (a peptide) due to the nature of the mechanism of non-covalent binding to the drug. Albeit, SNAC exposed the hydrophobic region of insulin and there were no changes in TJ proteins or membrane integrity, represented by an LDH assay and a mannitol transport assay in an insulin-modeled study [54]. There are also studies that contradict the carrier mechanism. A study on the application of SNAC to cromolyn sodium, a low-molecular-weight molecule with low absorption in the GI tract due to its high hydrophilicity, presented results conflicting with the carrier mechanism. When observed through the partitioning solvent system, the lipophilicity change in cromolyn sodium by SNAC was not observed, and SNAC was observed to increase the membrane fluidity of the Caco-2 cell monolayer [131]. The change in membrane fluidity was observed by the treatment with 83 mM SNAC, with the difference existing at the concentration levels higher than that of 33–55 mM SNAC used in the existing carrier mechanism study.

Oral semaglutide studies have suggested another mechanism of SNAC. Novo Nordisk presented a mechanism by which SNAC enhances oral absorption of semaglutide using various in vitro, in vivo, ex vivo, and clinical studies [61]. They put forth that the PE mechanism of SNAC works specifically for semaglutide. They found that SNAC protects semaglutide from enzymatic degradation in the stomach during uptake by elevating the local pH, and equally increases the transcellular uptake of semaglutide by the mechanism of induction of monomerization of semaglutide and increase in membrane fluidity (Figure 6). One of the major hurdles to the oral administration of peptide/protein drugs is the degradation by the pH environment and enzymes in the stomach. Therefore, an enteric coating is used commonly in many strategies utilizing PEs, to prevent the breakdown of the payload in the stomach and mainly target absorption in the small intestine [48,108,134,135]. In contrast, Novo Nordisk’s oral semaglutide, Rybelsus^®^, is a formulation designed to be absorbed after erosion of the entire tablet in the stomach, rather than applying an enteric coating technology. They conducted a study on the optimization of Rybelsus^®^ formulations through ex vivo experiments using dogs and gamma scintigraphic imaging of healthy adults [61,136]. They mentioned that the small intestine, which has a large surface area, is the main absorption site for drugs, disregarding the stomach as the absorption site for peptide/protein drugs. Nevertheless, they suggested that it is appropriate to target absorption in the stomach, as is the case with semaglutide, which must be co-formulated with a PE such as SNAC. Their strategy was to devise a formulation in which complete tablet erosion (CTE) occurs in the stomach by local disintegration by sinking to the lower part of the stomach through optimization of the formulation. CTE was confirmed in the stomach when taken with 240 mL of water on an empty stomach, and the average time to reach CTE was predicted to be about 1 h. Furthermore, clinical trials analyzed the absorption pattern of semaglutide according to the amount of water consumed. Regardless of the water intake, the Rybelsus tablet reached CTE in the stomach; however, it was found that when ingested with a relatively small amount of water (50 mL), the erosion time of the tablet increased, resulting in higher plasma semaglutide exposure [61,136]. Consequently, the mechanism of SNAC has been studied as a carrier mechanism based on a non-covalent bond, and a semaglutide-specific mechanism studied during the development of oral semaglutide. Therefore, it seems difficult to specify and define the general mechanism of SNAC as a PE.

Emisphere, which developed SNAC, tested it for vitamin B12, heparin, and insulin. After this technology was transferred to Novo Nordisk, numerous clinical trials on semaglutide have been conducted (Appendix A). In addition, SNAC was further applied to peptide drugs such as Novo Nordisk’s new GLP-a agonist (NNC0113-2023) and PCSK9 inhibitor (NNC0385-0434).

Numerous clinical trials conducted with semaglutide and SNAC for the successful development of Rybelsus^®^ can be broadly classified into three categories. First, the efficacy, safety, PK profiling, and dosage setting of semaglutide/SNAC. In terms of efficacy, a phase 3 trial targeting patients with T2DM managed only with diet and exercise (PIONEER 1) and patients with T2DM uncontrolled with metformin (PIONEER 2) demonstrated superiority. In PIONEER 1, oral administration of 3–14 mg of semaglutide caused a dose-dependent decrease in HbA_1c_ of 0.7–1.4%, and the highest dose of 14 mg also showed a significant weight loss in participants of 2.3 kg [137]. In PIONEER 2, semaglutide was compared with empagliflozin, an SGLT-2 inhibitor, which has a weight loss and hypoglycemic effect. When administered at 26 weeks, semaglutide showed superior HbA_1c_ reduction compared to empagliflozin, and at 52 weeks of administration, it showed superiority in both HbA_1c_ and weight loss [138]. For PK and safety, clinical trials were conducted not only on healthy subjects but also on renal [139] and hepatic [140] impaired subjects and patients with diseases in the upper gastrointestinal tract [141], which is the site of absorption of semaglutide. In all cases, there was no significant effect on the PK profile and safety concerns of semaglutide, and there was no specific safety concern due to the disease.

The second category is the effect on reciprocal PK with concomitant drugs. In particular, due to the nature of the indications of semaglutide for chronic diseases such as diabetes and obesity, the patient is likely to be on multiple drugs in combination. In addition, since PE is used, it is also necessary to investigate the effect of the PE on the drug used in combination. Thus far, 11 concomitant drugs have been investigated at the clinical trial stage. Among them, the PK of four concomitant drugs: ethynylestradiol/levonorgestrel, digoxin, lisinopril, and warfarin, were not affected by semaglutide/SNAC [142,143]. Conversely, there was an increase in exposure to rosuvastatin, furosemide, levothyroxine, and metformin with the combination of semaglutide/SNAC [142,143,144].

The third category is the study of formulations. Beyond optimizing the basic semaglutide/SNAC dose, Novo Nordisk has conducted clinical trials on combinations of other excipients (otherwise called helping agents). Although the results of the clinical trial were not disclosed, they suggested that the study of excipients added together with the oral formulation to which PE is applied is more important than that of conventional oral formulations. In effect, the two clinical trials conducted by Novo Nordisk aimed to examine the effect of an additional helping agent on the dosage strength of semaglutide. That is, compared to the classical excipient, it functions as a more active type of excipient that can help the complex action of peptide API and PE.

Riley et al. extensively studied the safety of SNAC at the in vivo level and in the course of Rybelsus’ clinical trials [145]. In vivo toxicity studies were conducted in a rat model for subchronic toxicity and peri- and post-natal developmental toxicity of SNAC. In Sprague Dawley rats (male (n = 20), female (n = 20)) given SNAC at 2 g/kg/day for 13 weeks, the mortality rate was 20% in males and 50% in females, showing a significant mortality rate, especially in females. Although the exact cause of death has not been elucidated, in comparison to the control group, changes in behavioral patterns (decreased activity, prostration, unkempt appearance, etc.) that persisted for several hours after SNAC administration; decreased globulin concentrations, especially in male rats; and a slight increase in liver and kidney weights were observed in the SNAC group. A sub-chronic toxicity study in Wistar rats was conducted with SNAC treatment at 0.1, 0.5, and 1 g/kg/day for 13 weeks. No death or clinical signs were observed at the highest dose of 1 g/kg/day, and they concluded that the no-adverse-effect level (NOAEL) for the Wistar rats was 1 g/kg/day. In the peri- and post-natal developmental toxicity study, oral administration of 1 g/kg/day from implantation to lactation of rats was conducted to evaluate the exposure and toxicity of SNAC in the uterus and breast milk of the offspring. There was no effect on the growth and development of surviving offspring exposed to SNAC, albeit that an increase in the number of stillbirths during pregnancy was observed [146]. Unlike MCFA, given that it does not exist in nature, SNAC may have more safety considerations. However, it has obtained FDA’s GRAS status due to the accumulation of preclinical and clinical trial data that have been conducted so far, and has been established as key in the development of oral semaglutides without serious side effects.

## 5. Discussion

Oral administration of peptides using PEs is a simpler approach compared to strategies using nanoparticles and microneedle devices. However, there are still some limitations to be solved and issues to be considered. The first concern is the safety, efficacy, and reproducibility of PEs. MCFA and SNAC have independent safety status, and through this it has been possible to overcome regulatory hurdles. However, many PEs still need more research regarding their safety. As with MCFA and SNAC, most PEs have no established mechanism of action. Therefore, a more extensive toxicological investigation such as the off-target action in the intestine and the entire body is required in addition to the safety concerns directly related to the presumed mechanism. Damage to the intestinal tissue and related intestinal inflammation are the main safety concerns in transcellular PEs caused by membrane perturbation. Whereas the absorption of unwanted substances (enterotoxin, etc.), changes in the TJ integrity of other tissues after systemic absorption, and off-target action by sub-signaling are emerging as representative safety concerns for paracellular PEs based on TJ modulation. Additionally, as the development trend in peptide drugs for metabolic diseases increases, continuous dosing is required. Accordingly, the effect of long-term administration of PEs on the intestinal normal flora has been pointed out.

It seems that simply combining peptide drugs and PEs does not yet yield sufficient efficacy. The efficacy of a PE is directly related to the required amount of the peptide drug, an API. The first example of this is oral semaglutide (Rybelsys^®^). The bioavailability of Rybelsys^®^ was 1% in a dog model [61]. In effect, the initial dose of injectable semaglutide (Wegovy^®^) administered once a week is 0.25 mg and the maintenance dose is 0.5–2 mg, whereas Rybelsus^®^ is administered with an initial dose of 3 mg and a maintenance dose of 7 or 14 mg per day. That is, the amount of semaglutide consumed in the oral drug is about 80 or more times higher than that in the injectable formulation. An example of this can also be found in oral octreotide (Mycapssa^®^) [147]. In a phase 1 study of Mycapssa^®^, an oral octreotide does of 20 mg showed equivalent PK parameters to the 0.1 mg dose of the subcutaneous injectable octreotide. When comparing the actual dose of Mycapssa^®^ with existing injectable products, 0.1 mg is administered 2–3 times a day for the subcutaneous injectable formulation, and 10–30 mg is injected once a month for the depot formulation. Conversely, Mycapssa^®^ requires 20 mg of octreotide twice a day. In this way, the PE is as important as the API in terms of safety and efficacy.

In addition to safety and efficacy, reproducibility is an important concept. This is because if the effect of enhancing the absorption of peptides by a PE is vulnerable to differences between individuals and dosing conditions within individuals, a decrease in the therapeutic effect or side effects may be induced. In effect, there were cases where it was not possible to derive the difference in the PK parameter according to the administered dose due to the large variability between subjects in the clinical trials [111]. Another study equally mentions the need to lower the variability between subjects in future research, with an SD value of 4% and an average value of relative oral bioavailability of 7% [127].

The second point to be considered is the construction of an experimental model and the design of the formulation considering the action of PEs in the dynamic environment of the actual gastrointestinal tract. The in vitro experiment using the Caco-2 monolayer—the center of PE research—is a static experimental model and does not represent the actual dynamic environment of the gastrointestinal tract. This leads to a poor prediction of the effect of PEs in subsequent in vivo and clinical studies, implying that the effect of PEs is practically negligible and may lead to the failure of oral peptide development. The two main causes of the discrepancy between the static experimental model and the actual gastrointestinal tract are considered to be dilution and temporal synchronization of the PE and peptide in the gastrointestinal tract. First, in the case of dilution, as the PE and peptide are eluted into the bulk fluid of the gastrointestinal tract, PE is diluted to a concentration below the threshold that enhances absorption, and the peptide to be absorbed is moved away from the absorption site. In Rybelsus^®^, targeting absorption in the stomach, the Rybelsus^®^ tablet is designed to localize and stay in the lower part of the stomach and CTE occurs. The design of the formulation appears to reduce the dilution of SNAC and semaglutide in the gastrointestinal tract. Novo Nordisk’s research team investigated the difference in absorption of semaglutide according to the amount of water consumed along with the drug in a clinical trial, and found that the AUC of semaglutide increased when taking it with a small amount of water [136]. They mentioned the possibility that higher concentrations of SNAC and semaglutide in the stomach facilitate absorption, as well as slowing erosion and increasing gastric retention in the stomach. This is actually reflected in dosing regimens; patients are guided to take the tablets with less than 4 ounces of water. Recently, as a method to ensure proper synchronization of the PE and peptide, and to prevent dilution, the development of formulations capable of uni-directional release, i.e., gastrointestinal patches [148,149] and microcontainers [150,151], is emerging. Whatever its shape, it releases the PE and peptide only in the direction of the mucous membrane of the gastrointestinal tract, preventing dilution by elution into the bulk fluid, ensuring their synchronization by simultaneously or sequentially eluting the PE and peptide.

The third consideration is the properties of the peptide drug to be considered for establishing a PE strategy. Foremost, the natural properties of the peptide itself can be a determinant of the degree of its absorption in the gastrointestinal tract. The first of these properties is its susceptibility to enzymes in the gastrointestinal tract. In addition to overcoming the epithelial barrier expected from the PE, the other big task to be solved is overcoming the enzymatic barrier. A study on the stability of various peptides in gastrointestinal fluid confirmed that the enzymatic stability differs greatly depending on the sequence and structure of the peptide. GnRH analogues (-relin), including leuprolide and goserelin, have resistance to pepsin due to the amino acid substituted with the D-form at position 6, and show good stability in gastric fluid (GF). On the contrary, it does not have a protective effect on other intestinal enzymes including trypsin and chymotrypsin, and it has been confirmed that it is rapidly decomposed in intestinal fluid (IF). Moreover, the cyclic peptides cyclosporine, desmopressin, and octreotide showed excellent stability compared to other peptides in both GF and IF [152]. Furthermore, the stability of these three cyclic peptides in the gastrointestinal tract seems to have contributed to the development of some oral products: Sadimmun^®^, Minirin^®^, and Mycapssa^®^. Consequently, in overcoming the enzymatic barrier, the properties of the peptide itself are important, and to improve this, chemical modifications such as amino acid substitution and cyclization can be useful.

The subsequent consideration is the effect of peptide charge and modifications such as acylation. This is related to the migration of the peptide to the epithelial cell membrane in the gastrointestinal tract. This migratory behavior will also affect the establishment of the PE strategy. The charge of the peptide will affect the migration to and behavior near the membrane. Therefore, a PE strategy considering these characteristics should be established. A study on PIP640, a peptide PE and a TJ modulator, investigated the absorption enhancement effect for exenatide and calcitonin, which have similar hydrodynamic sizes but pI values of 4.9 and 9.3, respectively. It was observed that positively charged calcitonin had more improved bioavailability than negatively charged exenatide after treatment with PIP640 [153]. This suggests that the properties of the peptide, such as charge, can affect the absorption enhancement effect caused by the PE. Changes in properties due to peptide modification are also major factors to consider. A typical modification is acylation, which is intended to increase the half-life through albumin binding. A study comparing absorption enhancement patterns by transcellular PE (SDS) and paracellular PE (EGTA) for GLP-2 and its acylation derivatives observed varying absorption patterns dependent on acylation and the length of its carbon chain. The amount of GLP-2 bound to the cell membrane or uptaken into the cytoplasm increased according to the increase in the carbon chain length (C8–C12–C16), and for C8 to C12, the amount permeated into the increased acceptor compartment also increased. However, when acylated with C16, a lower absorption compared to the control was found. In addition, when comparing the absorption-enhancing effect of GLP-2 acylated with C16 by treatment with SDS and EGTA separately, the absorption enhancement by SDS was found to be superior. This was not observed in the control group and the group acylated with a shorter carbon chain. Furthermore, the research team observed an increase in self-association between peptides by acylation, and argued that this could cause a change in the permeability pattern [154]. These results indicate the need for a PE strategy that considers the increased interaction with the membrane by acylation. A PE strategy that reflects this can be seen in the examples of semaglutide and SNAC. For semaglutide, an acylated peptide, a strategy for the transcellular pathway was used via SNAC. Equally, another strategy was monomerization of semaglutide by SNAC.

The fourth issue concerns the regulatory approval of oral peptides to which the PE strategy has been applied. Considering the flow of changes in the biowaiver criteria of API belonging to BCS Classes 1 and 3 of immediate release solid formulations, it can be seen that regulatory agencies are taking a more restrictive standard for excipients. According to the 2017 FDA guidance, the excipient of the drug belonging to BCS Class 1 would not affect the absorption of the API [155]. Exceptionally, it was specified that surfactants such as polysorbate and excipients such as sorbitol and mannitol, which increase the GI fluid volume to decrease the residence time in the small intestine and affect drug absorption, should not differ qualitatively and must be quantitatively similar. Then, in 2020, the International Council for Harmonisation of Technical Requirements for Pharmaceuticals for Human Use (ICH) published “M9 guideline on biopharmaceutics classification system-based biowaivers (Step 5)”, and the FDA published “M9 Biopharmaceutics Classification System-Based Biowaivers” reflecting this in the next year, which replaced the 2017 FDA guidance [156]. The difference from the 2017 guidance was the disappearance of the direct statement “The excipient has no effect on the absorption of APIs belonging to BCS Class 1*”*, and even in the case of BCS Class 1, expression of concern that excipients may affect the absorption of APIs was emphasized. Of course, since peptide drugs belong to BCS Class 3, quantitative and qualitative differences from the excipients of the reference product were strict from the beginning of the introduction of the biowaiver concept. However, this regulatory flow is an example that reflects the stricter regulations being applied to excipients, more specifically, absorption-modifying excipients [157]. Unlike classical excipients, PE is a substance added ‘intentionally’ with the more active purpose of enhancing the absorption of APIs. Therefore, there is no doubt that there will be higher regulatory hurdles on the existing concepts of excipients. In particular, in the case of PEs which have obtained the safety status of food additives, such as MCFA, and GRAS, such as SNAC, or PEs that have not been approved as an excipient by regulatory agencies, the corresponding authorization for a new chemical entity may be applied. Similarly, due to the characteristics of a PE, it may not be appropriate to apply the concept of a biowaiver in the existing BCS Class 3 APIs at the time of approval of the generic product of oral peptide pharmaceuticals to which the PE is applied.

A final point to consider is the economics of oral peptide pharmaceuticals. As in the examples of Rybelsus^®^ and Mycapssa^®^ mentioned above, the oral formulation of the peptide developed with the PE strategy reaches almost 100 times the amount of peptide consumed compared to the conventional injectable dosage form. This increases the production cost of oral dosage forms which is unreasonably high and thus they have a very low economic feasibility. However, it is premature to judge the production cost and economic feasibility of oral peptide pharmaceuticals simply by the amount of peptide used as an active ingredient. Novo Nordisk argued that Rybelsus^®^ had a higher API cost than Victoza^®^ (liraglutide injectable product), but that the unit cost was not significantly different when considering the delivery cost of the injectable formulation [158]. The delivery cost of the injectable formulation includes the device for injection and the aseptic conditioning process involved in its production. This raises an opportunity to consider the increase in unit cost due to the production of injectable products and the direction of the development of oral peptide pharmaceuticals, given that this is the unit cost evaluated in Rybelsus^®^ with oral bioavailability at approximately 1% level. If the oral bioavailability increases to a level greater than 1%, oral dosage forms can have a much lower unit cost compared to injectable dosage forms.

## 6. Conclusions

Oral macromolecule (peptides/proteins) delivery has been studied for decades, and a strategy using PE has recently yielded results at the peptide level. It was oral semaglutide that opened the door to oral peptide drug products. In particular, insulin, which has been a target of oral delivery since the beginning of the study of oral macromolecules, has not yet achieved clear results, despite numerous studies. Therefore, oral semaglutide by Novo Nordisk is evaluated to be more meaningful in the successful initiation of oral peptide drugs. The subsequent oral octreotide also became an example of a successful conversion to an oral dosage form because the existing dosage forms had obvious disadvantages, such as subcutaneous injectable formulations that required frequent administration and depot dosage forms that cause considerable pain during administration.

Despite the successful development history of Rybelsus^®^ and Mycapssa^®^, the strategy of using PEs still has many limitations. Firstly, in terms of safety and regulation, the practically available PEs are not diverse. This means that a specific approval procedure for PEs should be established through a more extensive and systematic safety study on chemical entities that can be used as PEs and regulatory scientific research on PEs. Secondly, there is a need for research on more effective formulations. This includes research on the development of potent PEs that can further increase the bioavailability of macromolecules and research on next-generation formulations, such as unidirectional drug release considering the actual intestinal environment. These studies will reduce the required amount of peptide as an API in the process of converting from an injectable formulation to an oral formulation and will ensure the justification for the development of an oral peptide drug from an economic point of view.

## Figures and Tables

**Figure 1 pharmaceuticals-15-01585-f001:**
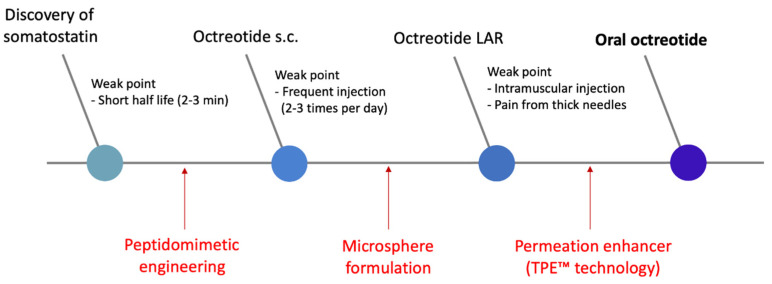
The historical chart of the octreotide formulation development process.

**Figure 2 pharmaceuticals-15-01585-f002:**
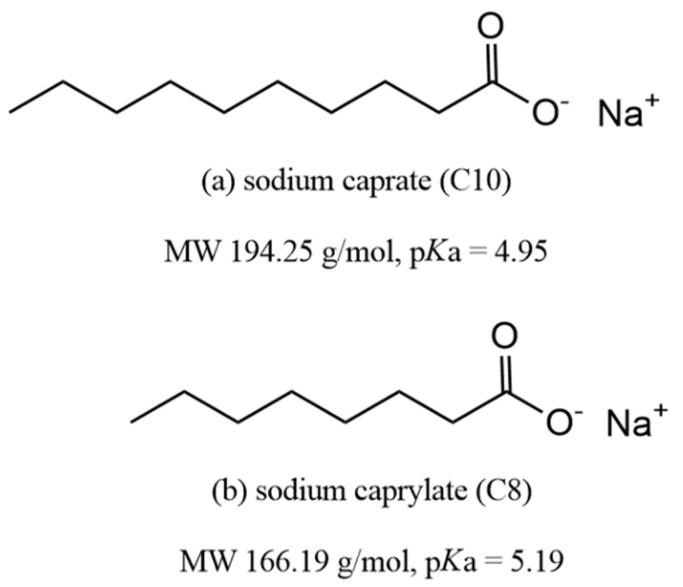
Structures of sodium caprate (C10) and sodium caprylate (C8).

**Figure 3 pharmaceuticals-15-01585-f003:**
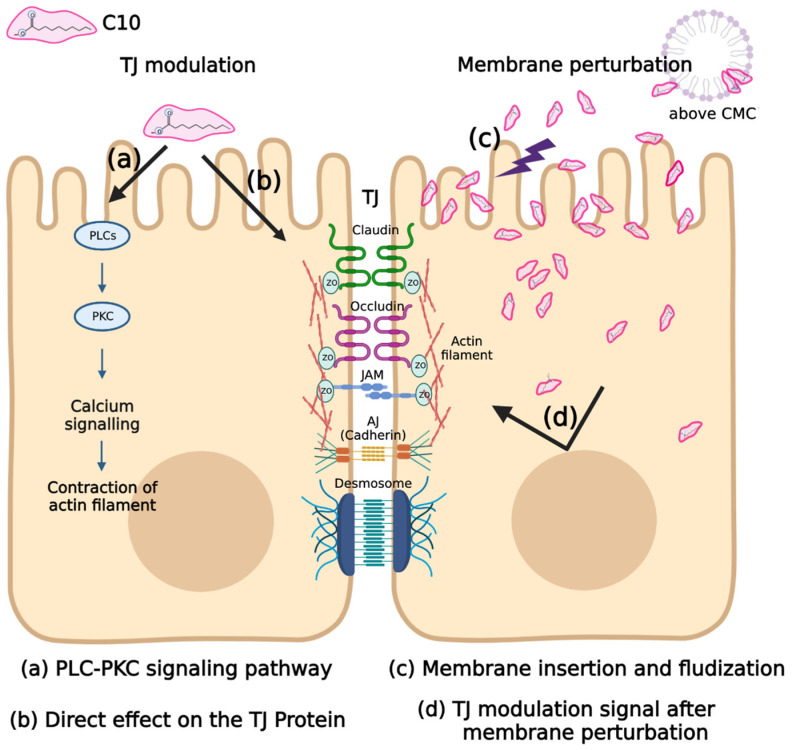
Hypotheses on the mechanism of C10. The diagram shows the simple classification of the mechanisms of C10 proposed to date. It is essentially divided into tight junction (TJ) modulation (left) and membrane perturbation mechanisms [68]. The TJ modulation hypothesis is divided into a Ca^2+^-dependent PLC-PKC pathway mechanism (**a**) and a Ca2+-independent TJ protein modulation (**b**). As per the membrane perturbation hypothesis, the mechanism of action of C10 can be its insertion into the membrane to fluidize the membrane and enhance absorption into the transcellular pathway (**c**), and an unknown TJ modulation signal due to membrane perturbation (**d**).

**Figure 4 pharmaceuticals-15-01585-f004:**
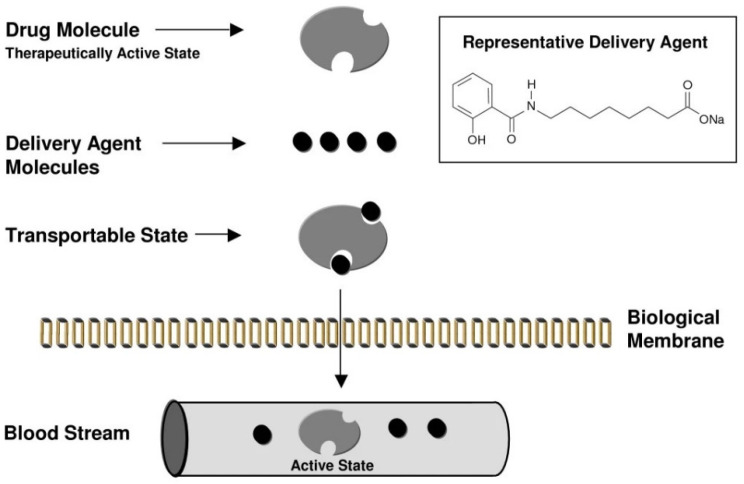
Schematic diagram of the mechanism of the Eligen™ technology (carrier mechanism). Emisphere researchers have argued that small carrier molecules with hydrophobic moieties increase lipophilicity through the formation of weak non-covalent bonds with drug molecules [120].

**Figure 5 pharmaceuticals-15-01585-f005:**
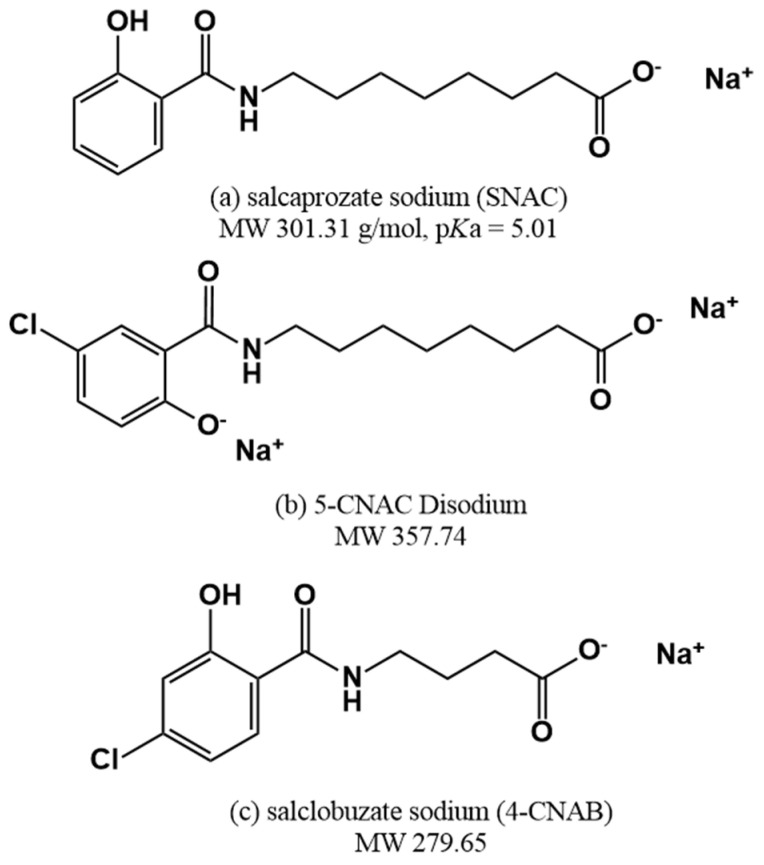
Structures of (**a**) SNAC, (**b**) 5-CNAC, and (**c**) 4-CNAB.

**Figure 6 pharmaceuticals-15-01585-f006:**
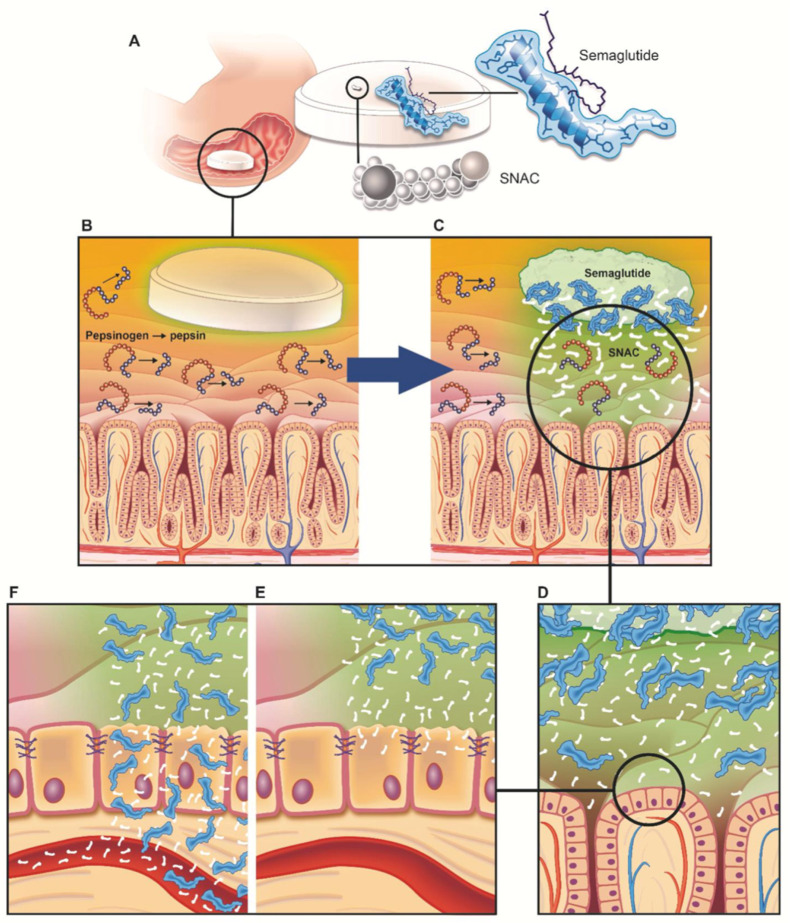
Formulation design of the Rybelsus^®^ tablet and schematic diagram of the semaglutide absorption enhancement mechanism of SNAC (published by the Novo Nordisk research team) [61]. The Rybelsus^®^ tablet (**A**) is completely eroded from the gastric mucosal surface (**B**,**C**) and SNAC increases the local pH to prevent the degradation of semaglutide from pepsin (**C**). Moreover, it indirectly weakens the self-association of semaglutide and helps to maintain the monomer state (**D**), and increases transcellular absorption by increasing membrane fluidity (**E**,**F**). Reprinted with permission from ref. [61]. Copyright 2018 *Science*.

**Table 1 pharmaceuticals-15-01585-t001:** Summary of chemical permeation enhancers.

Permeation Enhancer	Proposed Mechanism
MCFAs (C8, C10)	Paracellular - Direct or indirect tight junction modulation Transcellular - Membrane perturbation and fluidization
Eligen™ technology-based PE (SNAC, 4-CNAB, 5-CNAC)	Transcellular - Carrier mechanism - Local pH elevating and inhibition of gastric enzyme, peptide monomerization, membrane fluidization (SNAC-Semaglutide)
EDTA	Paracellular - Decreased tight junction integrity by chelation of Ca^2+^ [37,38]
Bile salt	Transcellular - Hydrophobic ion pairing, membrane fluidization [39,40,41]
Acyl-carnitine	Transcellular - Membrane perturbation [42] Paracellular - Tight junction modulation inferred from the decrease in TEER [43,44]
Alkyl-maltoside	Combined transcellular, paracellular action - A complex and colon-specific mechanism expected mild surfactant characteristics [32,45]
Sodium docusate	Transcellular - Hydrophobic ion pairing [46]
Sucrose laurate	Combined transcellular, paracellular action - Tight junction opening after membrane fluidization [47]
Choline geranate (CAGE, Ionic liquid)	Paracellular - Tight junction opening, protection from enzymatic degradation, decrease in mucus viscosity [48]

**Table 2 pharmaceuticals-15-01585-t002:** Studies on the permeation enhancement ability of sodium caprate (C10).

Peptide (MW)	Experimental Model	Dosing	Enhancement Ratio	Reference
In vitro assay				
D-decapeptide (~1.2 kDa)	Caco-2 cell monolayer	20–25 mM	~7	[82]
Ile-Pro-Pro (325 Da) Leu-Lys-Pro (389 Da)	Caco-2 cell monolayer	5 mM	~2.5 (Ile-Pro-Pro) ~2 (Leu-Lys-Pro)	[83]
Cyclopeptide, (EMD121974) (589 Da)	Caco-2 cell monolayer	10 mM	10.6	[84]
Vasopressin (1.2 kDa)	Caco-2 cell monolayer	13 mM	10	[85]
Recombinant human epidermal growth factor (6 kDa)	Caco-2 cell monolayer	1% (50 mM)	10.6	[86]
**Ex vivo/in vivo assay**				
D-decapeptide (~1.2 kDa)	Rat, ileal instillation	0.5 mmol/kg	~5	[82]
Ebiratide (996 Da)	Rat, Ussing chamber (jejunum and colon)	20 mM	1.50 (jejunum) 3.84 (colon)	[87]
Enalaprilat (349 Da)	Rat, single-pass intestinal perfusion	10 mg/mL	9	[88]
Hexarelin (887 Da)	Rat, single-pass intestinal perfusion	5~20 mg/mL	Not enhanced	[88]
Insulin (5.8 kDa)	Rat, Ussing chamber (jejunum and colon)	20 mM	0.97 (jejunum) 2.50 (Colon)	[89]
Horseradish peroxidase (45 kDa)	Human, Ussing chamber (Colon)	10 mM	~2	[90]
Insulin (5.8 kDa)	Rat, loop administration	1% (50 mM)	Duodenum: not enhanced * Jejunum: enhanced * Ileum: 1.67 * Colon: 9 *	[91]
Salmon calcitonin (3.4 kDa)	Rat, colonal instillation	0.1% (5 mM)	Enhanced *	[92]
Insulin (5.8 kDa)	Rat, rectal infusion	50 mM	24.31 *	[93]
DMP728 (657 Da)	Rat and dog, oral administration	Rat (8 mg/kg) Dog (2 mg/kg)	2.70 (Rat) 1.36 (Dog)	[94]
Insulin (5.8 kDa)	Rat, oral administration	0.5% (25 mM)	3.79*	[95]
Elcatonin (3.4 kDa)	Rabbit, rectal suppository	30 mg	1.61*	[96]

* Pharmacodynamic results.

**Table 3 pharmaceuticals-15-01585-t003:** Studies on the permeation enhancement mechanism of sodium caprate (C10).

Experimental Model	Proposed Mechanism of Action (Rationale)	Evidence	Reference
Caco-2 cell monolayer	TJ modulation	*TEM* - TJ dilatation was observed after treatment with 13 mM C10. *Fluorescence microscopy* - It is observed that the peri-junctional F actin ring disbands over time, which corresponds to the permeation enhancement timeline of ^14^C-Mannitol.	[97]
Caco-2 cell monolayer	TJ modulation	*TEM* - TJ dilatation was observed after treatment with 10 mM C10. - Frequency of dilatations after exposure to C10 was more than 40% (much higher than C8 and C12). *Fluorescence microscopy* - F actin rings were shown to be disbanded after exposure to C10.	[74]
Caco-2 cell monolayer	TJ modulation (PLC activation and CaM-dependent contraction of actin filament)	*Intracellular Ca^2+^ measurement (Fluorometric Ca^2+^ analyzer)* - C10 increased the intracellular Ca^2+^ regardless of the presence or absence of extracellular Ca^2+^ (0.05%–0.25% of C10, especially at 0.25%). *Inhibitor treatment for PLC-CaM signaling* - Treatment with KN-62 reduced the permeation-enhancing effect of C10 on FD-4K. - Treatment with KN-62 and W7 inhibited the C10-induced TEER reduction effect.	[38]
Ex vivo Ussing chamber (Rat ileum)	TJ modulation	*TEM* - C10 (10 mM) caused dilatations in 34% of the visualized TJ regions. *Comparison of patterns with Cytochalasin B* - Dose–response curve (C10: 2.5–10 mM, CytB: 60–300 µM) for transepithelial potential difference and [^51^Cr]EDTA P_app_ are similar.	[99]
Human airway epithelial cell	TJ modulation (Ca^2+^-independent mechanism and direct effect on the TJ protein)	*Fluo-4 Ca^2+^ assay* - 30 mM C10 induced a rapid increase in Ca^2+^ from ER and returned to control levels by 300 s after exposure. *The effect of BAPTP-AM, La^3+^, and thapsigargin* - The addition of BAPTP-AM (intracellular Ca^2+^ chelator) did not affect the decrease in the TEER value of C10. - The addition of La^3+^ (membrane Ca^2+^-ATPase inhibitor) and thapsigargin (ER Ca^2+^-ATPase inhibitor) that maintained an increase in intracellular Ca^2+^, did not affect the decrease in the R_T_ value of C10. *The effect of signaling inhibitor* - Each treatment of U7, 48/80, H7, W7, and KN62 did not affect the reduction of the R_T_ value of C10. *Immunofluorescent Labeling and Confocal microscopy* - JAM and actin were clearly redistributed after exposure to C10 (ZO-1 did not change). - The redistribution was not blocked by BAPTA-AM. - Claudin-1 and -4 were redistributed immediately after C10 treatment.	[100]
- MDCK cell monolayer - Lipid raft isolation study	Membrane perturbation (Lipid raft disruption) TJ modulation (Displacement of specific TJ proteins)	*Western-blot analysis* - Claudin -4,-5, and occludin were displaced from the lipid raft.	[101]
- HEK-293 Cell expressing claudin-5-YFP, MDCK-2-Cell expressing Flag-claudin-5 - Ex vivo mouse brain capillary	TJ modulation (by reducing the membranous claudin-5 amount and the F-actin content)	*Immunofluorescent Labeling and Confocal microscopy* - Claudin-5-YFP homophilic interaction was lost and fragmented by treatment with 5, 7.5, and 10 mM C10 for 20 min, and recovered after removal of C10. - Treatment with C10 yielded no significant change in CellMask, a membrane-inserting dye. (Compared to MβCD, which is a membrane disruptor.) *Immunoblotting* - 7.5 mM C10 reduced membranous claudin-5 and intracellular F-actin in TJ-containing MDCK-2-Cells and in brain endothelial cells. (There was no change in ZO-1.) - Claudin-5 was found to be displaced from the triton-insoluble (lipids) fraction of MDCK cells after treatment with C10.	[102]
- HT-29/B6 Cell	TJ modulation (by reversible removal of tricellulin from the tricellular TJ)	*Two-path impedance spectroscopy* - A significant decrease in paracellular resistance was observed by C10, although transcellular resistance was not significantly changed. *Immunofluorescent Labeling and Confocal microscopy* - C10 specifically induced a decrease in tricellulin and claudin-5 signals, which was reversible after washout. *Localization of Sulfo-NHS-SS-biotin Permeation sites* - Biotin signal was detected only within or below tricellular cell contacts (not detected in bicellular cell contacts).	[103]
- Caco-2 cell monolayer - Ex vivo Ussing chamber (Rat colon) - Rat intestinal instillation	Membrane perturbation	*Quantitative real-time PCR and gene expression microarrays* - After exposure to 8.5 mM C10 for 60 min, IL-8, an inflammatory signal, increased 11-fold and 26-fold at 1 and 4 h of recovery, respectively, and then decreased to the control level after 24 h. *Attenuation Effect of misoprostol* - Pre-exposure of monolayers to the misoprostol (10 and 100 nM) for 30 min prior to 8.5 mM C10 addition for 60 min significantly attenuated C10′s capacity to reduce TEER and increase the [^14^C]-mannitol and FD-4K P_app_. - The protective effect of pre-incubation with misoprostol against C10 was detected by TEM. - In a rat colonic loop instillation experiment, misoprostol reduced the mean AUC and C_max_ of FD4K by 24% and 33%, respectively, compared with the results treatment with C10 alone. - SC51322 (EP_1_ receptor antagonist) negated the effect of misoprostol in preventing the C10-induced changes in the intracellular Ca^2+^, mitochondrial membrane potential, and plasma membrane permeability.	[104]
Caco-2 cell monolayer	Membrane perturbation (initial and fundamental mechanism) TJ modulation (by intracellular pathway arising from initial plasma membrane perturbation)	*Immunofluorescence of TJ proteins* - At 5 mM or higher C10, ZO-1 was internalized, and claudin-5 and occludin were also relocated and internalized. *Cytotoxicity assay* - LDH-Glo™, CellTox Green™, Neutral Red, and JC-1 assay results showed a decrease in the integrity of the plasma, nuclear, and mitochondrial membrane that was concentration-dependent (usually at 5 mM or higher) of C10. (This pattern was not observed in SNAC.)	[105]
- CG-MD simulation - US simulation - TIRF microscopy (FRAP analysis)	Membrane perturbation (insertion of C10 into membrane and transmembrane perturbation)	*CG-MD simulation* - When 100 mM C10 in the fluid composition of the fasting state was applied to the POPC membrane, it was inserted into the membrane at a level of 70–80% in 6 µs. *US simulation (PMF profile)* - In a POPC bilayer with a thickness of 4.05 nm, C10 had an energy minima at a distance of 1.46 nm from the membrane center. (Energy minima represent the maximum probability of finding the molecule.) *TIRF Microscopy (FRAP analysis)* - In the POPC-C10 mixed membrane composed of various concentrations of C10, the membrane diffusivity increased in a C10 concentration-dependent manner. - When the pure POPC membrane was treated with 100 mM C10, diffusivity was increased, and it was observed that the generated bleached holes were recovered after 60 s when C10 was removed.	[72]

TEM, transmission electron microscope; TJ, tight junction; PLC, phospholipase C; CaM, calmodulin; FD, FITC-dextran; TEER, transepithelial electrical resistance; ER, endoplasmic reticulum; P_app_, apparent permeability; JAM, junctional adhesion molecule; YFP, yellow fluorescence protein; ZO-1, zonula occludens-1; EP_1_ receptor, prostaglandin E2 receptor 1; POPC, 1-Palmitoyl-2-oleoylphosphatidylcholine; CG-MD, coarse-grained molecular dynamics; US, umbrella sampling; TIRF, total internal reflection fluorescence; PMF, potential of mean force; FRAP, fluorescent recovery after photobleaching.

**Table 4 pharmaceuticals-15-01585-t004:** Studies on the permeation enhancement of SNAC.

Peptide (MW)	Model	Dosing	Enhancement Ratio	Reference
Insulin (5808 Da)	Caco-2 cell monolayer	5 mM	~10	[54]
Semaglutide (4114 Da)	NCI-N87 cell monolayer	80 mM	~7	[61]
Octreotide (1019 Da)	Ex vivo Ussing chamber of rat (colon, ileum, upper jejunum, duodenum, and stomach) and human (colon)	20 mM 40 mM	Rat (20 mM): 1.4~3.4 Human: 1.5 (20 mM), 2.1 (40 mM)	[128]
SHR-2042 (GLP-1RA) (~4.5 kDa)	Rat, duodenal perfusion	0.6 g/kg 1.2 g/kg	7.8 (0.6 g/kg) 69 (1.2 g.kg)	[132]

**Table 5 pharmaceuticals-15-01585-t005:** Studies on the permeation enhancement mechanism of SNAC.

Experimental Method and Model Drugs	Proposed Mechanism of Action	Evidence	Reference
Fluorescence microscopy, Heparin	The transcellular pathway that does not involve membrane permeabilization and does not appear to be endocytosis.	*Fluorescence and confocal microscopy in the Caco-2 monolayer* - Uptake of ALEXA FLUOR™ 488 labeled heparin was observed in over 33 mM SNAC, which was present in the cytoplasm and nucleus (endocytic vesicles were not observed). - Permeation of ALEXA FLUOR™ 488 hydrazide, a fluorescent probe, and YOYO-1, a membrane-impermeable DNA dye, was not observed when treated with 33 mM SNAC. - During actin staining of the apical monolayer, no change was observed until treatment with 66 mM SNAC.	[130]
Voltage clamp method, 6-Carboxy-fluorescein (6-CF)	Transcellular pathway	*Voltage clamp experiment* - 10 mM EDTA increased the paracellular flux rate of 6-CF by 3.1-fold and the transcellular flux rate by 1.3-fold, whereas 33 mM SNAC caused a 1.6-fold and 7.2-fold increase, respectively. *TEER measurement and mannitol transport assay* - There were no significant changes in TEER and Mannitol P_app_ in 33–66 mM SNAC, showing an increase in the permeability of 6-CF.	[133]
Fluorescence microscopy, Insulin	Increased lipophilicity by non-covalent binding and the resulting transcellular pathway	*Fluorescence and confocal microscopy in Caco-2 monolayer* - Fluorescently labeled insulin was detected in the cytoplasm and along the plasma membrane. - Treatment with 55 mM SNAC/80 µM insulin did not change the distribution of the peri-junctional ring and occludin.	[54]
The standard shake-flask method and steady-state fluorescence emission anisotropy, Cromolyn sodium	Increased membrane fluidity, but not increased lipophilicity in cromolyn sodium	*The standard shake-flask method* - All three n-Octanol-, chloroform-, and PGDP-/water systems showed differences in cromolyn sodium concentration in the aqueous layer in the presence and absence of SNAC, albeit without statistical significance. *Steady-state fluorescence emission anisotropy* - 83 mM SNAC alone or mixed with 10 mM cromolyn sodium significantly increased the fluidity of the core hydrophobic region as well as the surface polar region of Caco-2 cell membranes (not observed in 10 mM and 63 mM SNAC).	[131]
Various in vitro, in vivo/ex vivo assays	Protection against enzymatic degradation via local buffering actions and semaglutide-specific transcellular absorption in the stomach	*For transcellular mechanism* - (In vitro) In the NCI-N87 cell monolayer, both SNAC and EDTA increased the P_app_ of semaglutide similarly, but SNAC-induced intracellular accumulation of semaglutide and EDTA did not. - (In vitro) SNAC yielded a gradual reduction in the main transition temperature (T_m_) of DMPC membrane. - (Ex vivo) As a result of immunofluorescence. and TEM imaging of semaglutide immunoreactivity, semaglutide was present in the intracellular region, especially in the cytoplasm, via SNAC. *For semaglutide and SNAC specificity* - (In vivo) SNAC did not enhance the absorption of liraglutide, and the *ortho*-isomer of SNAC did not enhance the absorption of semaglutide. *For local buffering action* - (In vitro) Semaglutide/SNAC tablets augmented the pH of SGF from acidic to neutral within 5 to 15 min, the semaglutide tablet without SNAC had no apparent effect on the pH. - (In vitro) In the stability test of semaglutide in solutions of various pH values, including pepsin, t_1/2_ at pH 2.6 was 16 min, t_1/2_ at pH 5.0 was 34 min, and at pH 7.4, almost no degradation was observed over 50 min. *For absorption in the stomach* - (In vivo) As a result of gamma scintigraphic imaging, it was confirmed that complete erosion of the semaglutide/SNAC tablet was achieved in the human stomach in an average of 57 min. - (In vivo) Plasma concentrations of semaglutide were comparable in pyloric ligated and non-ligated dogs.	[61]
- CG-MD simulation - US simulation - TIRF microscopy (FRAP analysis)	SNAC does not appear to exhibit a transcellular mechanism by membrane insertion and perturbation.	*CG-MD simulation* - When 100 mM SNAC in the fluid composition of the fasting state was applied to the POPC membrane, fewer molecules of SNAC than those of C10 and C8 were adsorbed onto and incorporated into the membrane surface. *US simulation (PMF profile)* - In a POPC bilayer with a thickness of 4.05 nm, SNAC had an energy minima at a distance of 1.93 nm from the membrane center (energy minima represent the maximum probability of finding the molecule). *TIRF Microscopy (FRAP analysis)* - In the POPC–SNAC mixed membrane composed of various concentrations of SNAC, the membrane diffusivity decreased in a SNAC concentration-dependent manner.	[72]

PGDP, propylene glycol dipelargonate; DMPC, 1,2-dimyristoyl-*sn*-glycero-3-phosphocholine; SGF, simulated gastric fluid.

## Data Availability

Data is contained within the article and Appendix A.

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
