# Peer review of "Gastrointestinal Permeation Enhancers for the Development of Oral Peptide Pharmaceuticals"

_pharmaceuticals, 2022, doi:10.3390/ph15121585_

Round 1

Reviewer 1 Report

In this paper, the permeation enhancers for oral peptide were reviewed. The mechanisms of MCFAs (Medium chain fatty acids) and Eligen™ Technology-based PE penetration promoters were explored and summarized. The content of the article is meaningful and  attractive, some suggestions were listed below:

1.    1. Some permeation enhancers were also well explored in recent years, the summary and discussion of the characteristics of permeation enhancers would enhance the comprehensiveness of the article.

2. The historical development chart of octreotide formulation development process would be more simple and easy to understand.

3.The classification of permeation enhancer should be unified, the Permeation Enhancer – Eligen™ Technology-based PE (line 408)” was based on the technology , but “3. Permeation Enhancers – MCFAs (Medium chain fatty acids) (line 158)” was classfied based on the molecular structure.

4. The Permeation Enhancing effect was accurately also affected by the properties of  biomacromolecules and the interactions beteween permeation enhancers and drugs, which should be further discussed.

5. The formulation developments and applications of permeation enhancers in research should be also reviewed.

Author Response

We appreciate the reviewer for carefully reviewing our manuscript and providing constructive critiques. The response to the reviewer 1 comments  was attached. Please find it.

Reviewer 2 Report

The article is well presented and study is novel. Minor english language settings are required

Author Response

The authors thank you very much for this positive comment. English of the manuscript was edited by a professional, native English-speaking editor at Wordvice (Essayreview). Please see the attched file.

Reviewer 3 Report

The authors' review discusses permeation enhancers, molecules that facilitate the delivery of peptides through the intestinal epithelium. This is important because permeation enhancers can in turn allow for the oral administration of peptides, something that remain both challenging but highly valuable. 

My concerns about this review are listed below.

1) the writing is generally awkward and it is difficult to read this manuscript. There are too many English language issues for me to list them all herein. I would point to general trends that should be corrected: 1) very long sentences "xxx and xxx and xxx", 2) redundancy in words ("studies") and in ideas, 3) unusual way to break paragraphs (I wouldn't start a paragraph with "as such" because it clearly shows that the same idea is still being discussed, 4) logic ("the first concern is the safety, efficacy, and reproducibility"...that's 3 concerns...and later "in addition to safety and efficacy, an additional important concept is reproducibility"...redundancy).

2) The second main issue I have is about the "mechanisms" described. The figures provided are not very informative and I would not refer to them as representative of mechanisms. They highlight potential targets and potential effects...but they do not illustrate how these effects take place. 

Maybe it would be beneficial to provide models for how the PE can work (with a bit more molecular information), even if this is speculative.

Reviewer 4 Report

The manuscript "Gastrointestinal Permeation Enhancers for the Development of Oral Peptide Pharmaceuticals" went through two major technologies to enhance GI permeability. The manuscript is of high quality and should be published in Pharmaceuticals. It will enable the readers to learn about two success stories of getting peptides through oral delivery.

Author Response

Response to Reviewer 4 Comments

The manuscript "Gastrointestinal Permeation Enhancers for the Development of Oral Peptide Pharmaceuticals" went through two major technologies to enhance GI permeability. The manuscript is of high quality and should be published in Pharmaceuticals. It will enable the readers to learn about two success stories of getting peptides through oral delivery.

[Response]

We sincerely appreciate such a positive and constructive comment.